# Understanding the Antecedents and Effects of mHealth App Use in Pandemics: A Sequential Mixed-Method Investigation

**DOI:** 10.3390/ijerph20010834

**Published:** 2023-01-02

**Authors:** Xiaoling Jin, Zhangshuai Yuan, Zhongyun Zhou

**Affiliations:** 1Management School, Shanghai University, Shanghai 200444, China; 2School of Economics and Management, Tongji University, Shanghai 200092, China

**Keywords:** mHealth app use, psychological empowerment, COVID-19, event novelty, event criticality, even disruption

## Abstract

Pandemics such as COVID-19 pose serious threats to public health and disrupt the established systems for obtaining healthcare services. Mobile health (mHealth) apps serve the general public as a potential method for coping with these exogenous challenges. However, prior research has rarely discussed the antecedents and effects of mHealth apps and their use as a coping method during pandemics. Based on the technology acceptance model, empowerment theory, and event theory, we developed a research model to examine the antecedents (technology characteristics and event strength) and effects (psychological empowerment) of mHealth apps and their use. We tested this research model through a sequential mixed-method investigation. First, a quantitative study based on 402 Chinese mHealth users who used the apps during the COVID-19 pandemic was conducted to validate the theoretical model. A follow-up qualitative study of 191 online articles and reviews on mHealth during the COVID-19 pandemic was conducted to cross-validate the results and explain the unsupported findings of the quantitative study. The results show that (1) the mHealth app characteristics (perceived usefulness and perceived ease of use) positively affect mHealth app use; (2) mHealth app use positively affects the psychological empowerment of mHealth users; and (3) the characteristics of pandemic events (event criticality and event disruption) have positive moderating effects on the relationship between mHealth app characteristics and mHealth app use. This study explains the role of mHealth apps in the COVID-19 pandemic on the micro-level, which has implications for the ways in which mHealth apps are used in response to public pandemics.

## 1. Introduction

The continuous spread of the COVID-19 pandemic has placed pressure on healthcare resources, which has led patients to seek new solutions to health-related problems [1]. Due to the rapid spread of mobile devices worldwide, mobile health (mHealth) has great potential to enhance accessibility to specialist clinical diagnostics and treatment advice [2,3]. mHealth, providing health-related services through mobile devices and applications, offered an accessible and cost-effective solution to the pressures faced during the COVID-19 pandemic. There are various approaches to mHealth, such as text messaging, patient monitoring, and mobile telemedicine. These approaches can be deployed for health monitoring, health promotion, enhancing awareness, providing support at the point of care, and supporting decision-making [4,5,6]. These roles of mHealth are also reflected in the dramatic increase in mHealth app downloads that occurred during the global pandemic. In 2020, the number of mHealth apps available to iOS users continued to grow, and in the first quarter of 2021, it reached its peak at almost 54 thousand apps. Although mHealth is growing rapidly, practical problems, such as low user use and a lack of visible effects, remain. Therefore, it is important to understand the antecedents and effects of mHealth apps and their use during pandemics.

Prior studies have discussed the specific mechanisms of use intention and behavior in depth [7,8]. However, these studies still had some limitations. First, prior studies have adopted traditional technology adoption theories, such as the technology acceptance model (TAM), extended unified theory of acceptance, and use of technology (UTAUT), and motivation theory, to investigate the antecedents of mHealth app use (e.g., [9]). However, these studies did not consider the effects of major exogenous events (e.g., COVID-19). It is important to understand the mechanisms of mHealth app use during major events in order to empower users to navigate this technology [1,10]. Secondly, previous studies have examined the impacts of mHealth app use on health behavior and health status [11], but they have often ignored the psychological consequences of mHealth app use [12]. The critical role of mHealth apps is to support the “patient-centered” model of proactive care, which is widely considered to be closely related to psychological empowerment [13], defined as an individual’s psychological state and overall feeling of mastery over issues of concern [14]. Here, we consider the concept that enhancing citizens’ psychological empowerment in regard to health management is an important means of building resilience. Prior research on psychological empowerment has developed and applied this concept in various contexts, but little research has specifically discussed the issue of empowerment in the domain of mHealth [15]. Moreover, prior mHealth studies mainly explained the consequences of the psychological empowerment of mHealth users [16], lacking explorations of the mechanisms through which the psychological empowerment of mHealth users is formed. Hence, this study aims to address the following research questions:


*(1) How does event strength moderate the relationship between the technological characteristics and use of mHealth apps?*



*(2) How does mHealth app use affect the psychological empowerment of mHealth users?*


We applied a sequential mixed-method investigation to address the above research questions [17]. This mixed study included a questionnaire survey of mHealth app users and a complementary qualitative analysis of articles and reviews of mHealth apps in China. This mixed-method design enabled us to derive complementary holistic meta-inferences regarding mHealth app use in pandemics. We first used a quantitative method to develop and validate the research model and hypotheses. With respect to the antecedents of mHealth app use, we explored the impacts of technological characteristics (perceived usefulness and perceived ease of use) on users’ mHealth app use and explored the moderating roles of event features (i.e., event novelty, event criticality, and event disruption). In addition, turning to the impacts of mHealth app use, we explored the effect of mHealth app use on user psychological empowerment. Finally, we tested our proposed hypotheses and provided explanations for the unsupported findings using a qualitative design. This study provides the following theoretical contributions. Firstly, empowerment was introduced into the context of the mHealth app, and a definition of mHealth app empowerment was constructed. Secondly, it revealed the entire process of mHealth app use and explored the specific process mechanism. Finally, it innovatively incorporated event strength into the use model, revealing the moderating effect of the pandemic event strength on mHealth app use and advancing the integration of use and event research.

## 2. Literature Review

### 2.1. Antecedent of mHealth Use 

The mHealth app is a medical application based on mobile terminals that mainly provides services such as medical searches, appointment bookings, the purchase of medicines, and verification of professional information. Specifically, mHealth aims to increase access to medical services and health-related information and improve people’s ability to cope with critical situations, such as COVID-19.

Research on the antecedents of mHealth use considers a broader set of influences based on the integration of accessibility and usefulness factors. The current exploration of the antecedents of mHealth use was conducted from perception perspectives [18,19] using theories such as TAM and UTAUT [18]. Perceived usefulness and perceived ease of use, two key antecedents of technology adoption in TAM, have also been demonstrated to be important determinants of digital technology use behavior [20]. Perceived usefulness refers to the capacity of digital technology and its adoption to respond to threats or challenges posed by external events [21]. In the context of this study, it is defined as the degree of the capacity of the mHealth app and its usage to respond to the threat posed by the COVID-19 outbreak event. Perceived ease of use refers to the degree to which users find it easy to use digital technology [22]. In the context of this study, it is defined as the extent to which the use of mHealth is perceived to be effortless. Therefore, this paper considers these two technology perception factors (i.e., perceived usefulness and perceived ease of use) as antecedents to assess mHealth app use.

In regard to the impact of COVID-19, we investigated the role of event strength in influencing users’ antecedents of mHealth use. The key characteristics of events are novelty, criticality, and disruption. Event novelty reflects the extent to which an event differs from existing or previous events, behaviors, and characteristics [20]. Event criticality reflects the importance or priority of the event to the entity in question [23]. The more critical the event is, the more time, effort, and resources the entity will need to invest in response to the event. Event disruption reflects the extent to which external events change and disrupt daily activities. The more unexpected the event is, the more personal changes will be required in order to respond to the event. Based on this, this paper uses event novelty, event criticality, and event disruption to assess the strength of the COVID-19 outbreak event.

### 2.2. Impact of mHealth App Use

Prior studies on the impact of mHealth use have focused on two outcome variables: changes in health behavior (self-management activities) and changes in health status (health promotion) [24]. In the case of self-management activities, the mHealth impact studies can be divided into three categories. The first category focuses on patients’ disease-management needs and examines their self-management activities in relation to specific diseases using mHealth [25]. The second category focuses on patients’ resource needs, examining how mHealth helps patients to access resources related to self-management, such as human resources and social support, which promote self-management activities [26]. The third category focuses on patients’ needs in the context of daily life, such as the use of mHealth apps to access information and adopt healthy lifestyles [25]. Health promotion is another important impact of mHealth app use, as it aims to improve the patient’s health status. Health promotion studies have focused on changes in the clinical indicators of patients’ various conditions [27].

However, prior studies have neglected the psychological impacts of mHealth use. The key to the potential of mHealth apps is their capacity to psychologically empower users to manage their health care [28]. Psychological empowerment refers to an individual’s psychological state or sense of mastery over the issue of concern [14]. Taken together, Akeel and Mundy (2015) defined psychological (patient) empowerment as patients’ degree of control over, and responsibility for, their health status or condition (self-control), together with their belief in their ability to participate in medical consultation and decision-making processes (self-efficacy). Defining empowerment in a specific context requires the specification of two main aspects: the source of empowerment and the task of empowerment. On this basis, this study defines psychological empowerment in the mHealth app context (mHealth app empowerment) as the use of the mHealth app to enhance the user’s self-control and self-efficacy in regard to health management activities and consequences [16].

## 3. Research Model and Hypothesis

This study uses the antecedents of technology adoption (perceived usefulness and perceived ease of use) to represent the characteristics of mHealth apps. Furthermore, we draw upon event system theory to define the event strength of COVID-19 (novelty, criticality, and disruption). Figure 1 shows the research model. The main principles of the model include the following: (1) the impact of mHealth app characteristics on mHealth app use under the influence of pandemic events, and (2) the impact of mHealth app use on the psychological empowerment of mHealth users.

### 3.1. Effects of Technological Characteristics on mHealth App Use

The mHealth app technology characteristics have positive impacts on mHealth app use. Prior literature has demonstrated that the perceived usefulness and perceived ease of use of mHealth apps promote the use of this technology [29]. Wilson and Lankton found that perceived ease of use and perceived usefulness affect patients’ intention to adopt ehealth [30]. Following this line of research, we posit that the enhancement of mHealth app use is related to mHealth app characteristics, including their perceived usefulness and perceived ease of use. During the COVID-19 pandemic, users were receptive to mHealth apps if they felt that using them would be useful for maintaining health. Similarly, users accepted and used mHealth apps if they perceived that it would be easy to use this technology. Therefore, we propose the following hypothesis:

**H1:** *Perceived usefulness has a positive effect on mHealth app use*.

**H2:** *Perceived ease of use has a positive effect on mHealth app use*.

### 3.2. Effects of mHealth App Use on the Psychological Empowerment of mHealth Users

mHealth app use can enhance users’ sense of self-efficacy and self-control over their health status during the health care process [31]. Hence, we propose that mHealth app use positively influences mHealth app empowerment. Previous studies have confirmed that mHealth app use behaviors (e.g., intermittent use, continuous use) positively affect users’ perceptions of empowerment [32,33]. For example, Kang et al. (2017) found that patients feel empowered and are more willing to share knowledge in health care communities when using online health care platforms [29]. In the context of this study, by using mobile health apps for the purpose of healthcare participation during pandemics, users can improve their ability to obtain health information (health education), conduct doctor–patient communication (telemedicine), and maintain healthy habits (self-management). This improved ability and control can consequently enhance the psychological empowerment of mHealth users. Therefore, we propose the following hypothesis:

**H3:** *mHealth app use has a positive effect on the psychological empowerment of mHealth users*.

### 3.3. Effects of Event Strength on mHealth App Use

Events refer to life circumstances and experiences that shape an individual’s thoughts, feelings, and actions, which are usually closely related to their cognitive appraisal [34]. Event system theory holds that event strength has an indirect effect on individuals’ cognitive appraisals and their outcomes [34]. Event strength is considered to be an important factor affecting individuals’ use of mobile apps and their subsequent behavior. The key characteristics of events are novelty, criticality, and disruption [35].

Event novelty reflects the degree to which an event is unexpected, distinguishes it from existing and previous events, and requires individuals to adopt new behaviors in response to the event [20]. The outbreak of the COVID-19 pandemic has resulted in consequences such as social isolation and health risks [36]. These consequences threaten the ability of individuals to manage their health and raise concerns about the event in question. Concerns about the event often lead individuals to take measures in order to respond to it. When their health management ability is threatened, individuals are more receptive to measures that can enhance their health management ability and are more likely to gain a heightened perception of enhanced health management ability. Accordingly, the influences of mHealth app characteristics on mHealth app use will be strengthened. Specifically, with equal degrees of perceived usefulness or perceived ease of use, the novelty of a pandemic event will encourage users to accept mHealth app use in response to health threats. Therefore, we propose the following hypotheses:

**H4a:** *Event novelty positively moderates the effect of perceived usefulness on mHealth app use, and the effect is stronger when event novelty is high*.

**H4b:** *Event novelty positively moderates the effect of perceived ease of use on mHealth app use, and the effect is stronger when event novelty is high*.

Event criticality reflects the priority of the event and its important impacts on personal life, requiring significant time and effort in response [23]. The widespread and long-lasting impact of the COVID-19 pandemic, which remains a serious threat to people’s daily lives, has led to a special focus on, and special treatment of, the pandemic. This may have caused individuals to be more active in responding to the pandemic and eager to address the threat posed by the pandemic [37]. In such a case, individuals will be more receptive to mHealth apps and more likely to use them in order to address health threats. Specifically, with equal degrees of perceived usefulness or perceived ease of use, the criticality of a pandemic event will encourage users to accept the use of mHealth apps in response to health threats. Therefore, we propose the following hypotheses:

**H5a:** *Event criticality positively moderates the effect of perceived usefulness on mHealth app use, and the effect is stronger when event criticality is high*.

**H5b:** *Event criticality positively moderates the effect of perceived ease of use on mHealth app use, and the effect is stronger when event criticality is high*.

Event disruption refers to the degree to which an external event changes and causes interference and disruption in daily life, such as forced home quarantine [38]. Such events often diminish individuals’ ability to conduct themselves in a normal way, requiring additional analysis and behavioral changes in order to adapt to the event. Therefore, in a pandemic, mHealth app use is more readily accepted in response to health risks. Specifically, with equal degrees of perceived usefulness or perceived ease of use, the disruption caused by a pandemic will encourage users to accept mHealth app use in response to health threats. Therefore, we propose the following hypotheses:

**H6a:** *Event disruption positively moderates the effect of perceived usefulness on mHealth app use, and the effect is stronger when event disruption is high*.

**H6b:** *Event disruption positively moderates the effect of perceived ease of use on mHealth app use, and the effect is stronger when event disruption is high*.

## 4. Methodology and Results

We used a sequential mixed-method design, including a quantitative study (i.e., Study 1) and a follow-up qualitative study (i.e., Study 2) [17]. The mixed-method approach is suitable for addressing confirmatory and exploratory research questions and provides the potential to obtain stronger meta-inferences compared to a single method [39,40]. The mixed-method design is appropriate for this research for two reasons [41]. Firstly, a sequential design should be used if there is an existing theoretical basis for the study. In this study, the research variables were defined according to an established theory. Secondly, the mixed-method design helps researchers address the temporal evolution of the focal event (i.e., COVID-19) and can enhance the robustness of the results.

### 4.1. Study 1: Quantitative Study

#### 4.1.1. Data Collection

We conducted an online survey in China and collected data from mobile health app users in May 2020. We undertook a number of actions to ensure the quality of the survey process, including the back-translation approach when converting the English items into Chinese, pretests of the survey measures, the provision of incentives for participation in the survey, the use of IP checks to avoid duplicated responses, and the use of screening questions to prevent unqualified respondents, as well as tests for non-response bias, multicollinearity, and common method bias. After removing the data of those who failed the attention test (n = 70), answered faster than the time threshold (n = 76), or provided the same answer (n = 54), we obtained 402 valid responses. The demographic information of the valid respondents is provided in Table 1. A total of 42.5% of the questionnaire respondents were male, and 57.5% were female. In terms of age, most of the respondents were aged 18–45, accounting for 82.3%. Most respondents had a bachelor’s degree or higher qualification (72.4%). Most of the respondents had a monthly income of no more than 10,000 RMB, accounting for 87.1%. The demographics of our sample were consistent with those of previous studies involving Chinese mHealth app users [42].

#### 4.1.2. Measurements

The measures for all the constructs were adapted from previous studies and slightly modified to fit the research setting, and all the items were measured using a 7-point Likert scale. The questionnaire mainly included seven variables. The items for assessing mHealth app use were adapted from [43], and the items for assessing mHealth app empowerment were adapted from [44]. The items for assessing event disruption were adapted from [20], the items for assessing event criticality were adapted from [23], and the items for assessing event novelty were adapted from [34]. The items for assessing perceived usefulness were adapted from [21,22], and the items for assessing perceived ease of use were adapted from [17]. Since the reference scales were all in English, all the English items were translated into Chinese using the double-translation/back-translation method to ensure the consistency of the items of the questionnaire was retained in both Chinese and English. The specific questionnaire variables, question items, and sources are shown in Table A1. In addition to the main research variables, the questionnaire also included demographic variables (gender, age, education, income level) as control variables.

#### 4.1.3. Data Analysis and Results

Study 1 adopted SPSS and AMOS for the structural equation modeling. According to the confirmatory factor analysis, the fit indices of the measurement model (χ2/df = 2.14 < 3.00; RM-SEA = 0.05 < 0.08; CFI = 0.94 > 0.90; TLI = 0.93 > 0.90; NFI = 0.92 > 0.90) met the criteria. The measurement model was evaluated for its reliability, convergent validity, and discriminant validity. Reliability is best assessed by calculating Cronbach’s alpha and the composite reliability (CR), both of which should be above 0.70 [45]. As shown in Table 2, Cronbach’s alpha and the CR values for each construct exceeded 0.83, indicating that our constructs had appropriate reliability. Our constructs showed strong convergent validity, as the AVE of each construct exceeded the suggested threshold of 0.50, and the loading of each item exceeded the suggested threshold of 0.60 [45]. Additionally, as indicated in Table 3, our constructs showed a satisfactory discriminant validity.

After completing the measurement model test, the hypotheses were validated by structural model testing. The fit indices (χ^2^/df = 2.49 < 3.00; RM-SEA = 0.06 < 0.08; CFI = 0.93 > 0.90; TLI = 0.92 > 0.90; NFI = 0.91 > 0.90) indicated a favorable fit of the structural model with the data [46,47,48]. For the structural model, we used demographic variables, including gender, age, education, and income, as control variables that may affect the structural model to ensure the stability of the hypothesis test results.

The hypothesis test results are shown in Figure 2 and Table 4. Perceived usefulness (β = 0.60, t = 6.78, *p* < 0.001) and perceived ease of use (β = 0.22, t = 4.07, *p* < 0.001) both had significant positive effects on mHealth app use, explaining 46% of the variance in mHealth app use and thus providing support for H1 and H2, respectively. mHealth app use (β = 0.67, t = 9.85, *p* < 0.001) had a significant positive effect on mHealth app empowerment, explaining 31% of the variance in mHealth app empowerment, thus supporting H3. In regard to the moderating effects of event strength, our results showed that event novelty had no significant moderating effect on the positive relationships of perceived usefulness (β = −0.04, t = −0.54, *p* > 0.05) and perceived ease of use (β = −0.02, t = −0.35, *p* > 0.05) with mHealth app use, thus leading us to reject H4a–b. Event criticality had a positive moderating effect on the positive relationships of perceived usefulness (β = 0.17 t = 2.50, *p* < 0.05) and perceived ease of use (β = 0.11, t = 2.07, *p* < 0.05) with mHealth app use, supporting H5a and H5b. Event disruption had no significant moderating effect on the positive relationship between perceived usefulness (β = −0.06, t = −1.59, *p* > 0.05) and mHealth app use but had a significant moderating effect on the positive relationship between perceived ease of use (β = 0.11, t = 2.50, *p* < 0.05) and mHealth app empowerment. Thus, H6a was rejected, whereas H6b was supported. We used demographic variables, including age, gender, health status, and education, as control variables in the research model. The results showed that health status (β = 0.10, t = 2.17, *p* < 0.05) and education (β = −0.12, t = 2.18, *p* < 0.05) had significant effects on the psychological empowerment of mHealth users, whereas the other control variables (age and gender) had no significant effect.

We plotted the moderating effects to explain our findings from Study 1. As shown in Figure 3 and Figure 4, the sloped regression line indicating the effect of perceived usefulness or perceived ease of use on mHealth app use was positive and significant in terms of both high and low event criticality. However, mHealth app use increased more rapidly when perceived usefulness or perceived ease of use increased with high event criticality rather than low event criticality. As shown in Figure 5, the slope was positive and significant for both high and low event disruptions, and with high levels of event disruption, mHealth app use increased more rapidly than it did under low levels of event disruption, when the perceived ease of use increased.

### 4.2. Study 2: Qualitative Study

#### 4.2.1. Data Collection

In the qualitative study, we systematically conducted extensive online searches on well-known online social platforms with a large number of users, including Zhihu, WeChat Official Public Accounts, Sohu, and major app stores (e.g., Huawei Mobile Market, Apple Store). The search content mainly included articles (e.g., Zhihu blogs, WeChat Official Public Accounts tweets, Sohu news) and comments (major app stores) generated by users. For the search keywords, we used the Chinese names of the most well-known mobile health apps (e.g., Pingan Health App, Ding Xiang Doctor App, Miao Health) and combined them with terms related to the COVID-19 pandemic. The data collection period was from January 2020 to June 2020 during COVID-19. The formal data collection began on June 10 and ended on June 23, with a collection of 16,712 articles and reviews. After scrutinizing these articles and reviews, we removed 8963 articles and reviews that were identical due to cross-platform republication, 4509 articles or reviews that were not related to the pandemic, and 3049 reviews that were less than 10 words in length. This narrowed the dataset to 191 articles and comments, including 110 app store comments, 25 blogs from Zhihu.com, 20 tweets from WeChat official public accounts, and 36 popular articles from Sohu.com.

#### 4.2.2. Data Analysis

We used the open coding method to analyze the data [49], because this method allows the relevant information to emerge through the iterative process of examination and is widely used to develop themes and meanings [50]. Specifically, we grouped similar themes in the articles/comments into first-order concepts and categorized the similar first-order concepts into second-order themes [51]. The first-order concepts were statements extracted from the articles/reviews that reflected the main constructs in the research model and, hence, were relevant for understanding the hypotheses. The second-order themes described the relationships between the first-order concepts and provided evidence that either supported or did not support our hypotheses. For instance, one user comment stated: “During the pandemic, I was quarantined at home, bought medicine and saw a doctor with one touch, and did not realize before that I could also analyze my meal and sleep quality”. This comment reflects the idea that using the easy-to-use mHealth app (perceived ease of use) encourages people use the mHealth app more frequently (mHealth app use). Therefore, the comment can be coded into second-order themes, namely, “event disruption”, “perceived ease of use”, and “mHealth app use”, providing supporting evidence for H5b. To ensure the credibility of the qualitative results, we referred to Rao’s [52] test to assess the reliability and validity of the qualitative analyses. Before the formal coding, two researchers independently coded a random sample of 60 articles/comments. The inter-rater reliability of the coding was satisfactory, with a Cohen’s kappa value was 0.855, which was not significantly different, suggesting an acceptable level of agreement between the two coders. Additionally, the remaining articles/comments were randomly divided into two parts, each of which was coded independently by the coders. 

#### 4.2.3. Results

We tested the hypotheses according to hypotheticodeductive (H-D) logic [53]. Table A2 provides the first-order and second-order theme frequencies and interesting insights. We assessed the data related to each hypothesis and summarized the evidence indicating whether or not it supported the hypothesis based on the expression of the case. The expression of evidence in the table lists the representative cases related to each hypothesis and reveals the description and theoretical mechanism proving the hypothesis in the explanation column. 

As in the case of the quantitative study, the qualitative study also showed that technological characteristics (perceived usefulness and perceived ease of use) can enhance mHealth app use (H1 and H2). That is, use is promoted when the users believe that mHealth apps are usefulness and easy to use. Secondly, in line with the quantitative results, the qualitative study also provided evidence indicating the positive effect of mHealth app use on the psychological empowerment of mHealth users (H3). Finally, consistent with the results of the quantitative study, the qualitative study also provided evidence indicating that the positive effects of perceived usefulness and perceived ease of use on mHealth app use were strengthened by event criticality during COVID-19 (H5a and H5b). Additionally, the qualitative study also provided evidence indicating that the positive effect of perceived ease of use on mHealth app use was strengthened by event disruption during COVID-19 (H6b).

The qualitative study also allowed us to examine the relationship that was not supported in the quantitative study. Specifically, while the quantitative study showed that event novelty did not moderate the relationship between technological characteristics (perceived usefulness and perceived ease of use) and mHealth app use (H4a and H4b), the qualitative study provided a plausible explanation for this finding. As indicated by the representative evidence shown in Table A2, when users feel that it is useful and easy to manage their health using the mHealth app, this tends to promote greater use. This suggests that the positive relationship between the technological characteristics (perceived usefulness and perceived ease of use) and mHealth app use always exists and is not contingent on the degree of event novelty. Similarly, the quantitative study showed that event disruption did not moderate the relationship between perceived usefulness and mHealth app use (H6a).

Overall, the results of the qualitative study not only confirmed the results of the quantitative study but also supplemented and extended the quantitative study [42]. This was fully consistent with the objectives of triangulation and complementarity in mixed-method research. By integrating the results of the quantitative and qualitative studies, as shown in Table 5, we were able to draw interesting meta-inferences about the antecedents and impacts of mHealth app use during COVID-19 and provide some important theoretical and practical implications.

## 5. Conclusions

This research revealed an interesting set of findings that were found to be consistent in both studies. Firstly, the results of both studies confirm that mHealth app characteristics (perceived usefulness and perceived ease of use) have significant positive effects on mHealth app use and that mHealth app use has a significant positive effect on mHealth app empowerment. Secondly, both the quantitative and qualitative results confirm that the event strength of the COVID-19 pandemic (e.g., event criticality) positively moderated the positive effect of mHealth app characteristics on mHealth app use. However, some of the moderating effects were not confirmed in the quantitative study. In the case of these unexpected findings, the qualitative study provided complementary and nuanced insights. The possible reasons behind these findings may be related to the ways in which users perceive event strength and other influencing factors in qualitative studies. For example, the moderating effect of event novelty on the characteristics and use of mHealth apps was not confirmed. By comparing the mean and standard deviation of the event novelty, we identified a relatively high mean (5.99) with a small standard deviation (0.80). This indicates that the users did not differ significantly in their perception of the novelty of the COVID-19 pandemic event, which may have led to an insignificant moderating effect of event novelty.

### 5.1. Theoretical Implications

This study responded to the recent call for pandemic response research considering the use of mHealth apps during the COVID-19 pandemic. We propose that this research has the following theoretical implications. Firstly, this research enriched the mHealth literature by offering significant insights into the role of mHealth apps in the context of disruptive events (COVID-19). We showed that mHealth app use is an important causal mechanism linking the antecedents and effects of mHealth app use in pandemics. 

Secondly, we introduced psychological empowerment into the mHealth context and contextualized the definition of the psychological empowerment of mHealth app users. Our findings confirmed the positive effects of mHealth app use on the psychological empowerment of mHealth users. This finding enriches our understanding of the relationship between user usage and psychology and expands the literature on empowerment in the context of mHealth. 

Finally, our study contributed to event system theory by providing evidence supporting the relationships between the technological characteristics and use of mHealth apps. Prior studies on mHealth app use have focused on routine scenarios and rarely incorporated event strength into their examinations. Our findings confirmed that event characteristics exert moderating effects on technological characteristics in relation to mHealth app use. This paper thus extended the literature on the impacts of event characteristics on mHealth.

### 5.2. Practical Implications

This research has important implications for mHealth app stakeholders (e.g., health care providers, mHealth developers, and public administrators) with respect to the use of mHealth to meet the goal of improving mHealth app use during pandemics. First, this study revealed the specific action pathways of mHealth app use. Additionally, we provided guidance for users in regard to the use of mHealth apps (especially during the pandemic) and, consequently, empowered mHealth app users, boosting their confidence. Given the role of mHealth, health care providers could also consider accelerating the pace of digital transformation and providing more online healthcare services. 

Secondly, this study provides useful insights for mHealth developers regarding what technological characteristics are important for improving user’s mHealth use during COVID-19. Our results showed that one critical antecedent of mHealth use is the extent to which a mHealth app was perceived by users as an effective method in addressing health-care-related challenges caused by the pandemic. Thus, mHealth developer should focus on those technological features that strive to improve the quality of mHealth app design in terms of information, systems and services. 

Finally, this study demonstrated the important impacts of the characteristics of pandemic events on mHealth users in regard to their use of mHealth apps. For instance, the criticality and disruption of COVID-19 influenced the ways in which users sought to access healthcare services in hospitals offline. Thus, in the future, public administrators should judge the COVID-19 situation to encourage the public to use digital tools so as to manage their health.

### 5.3. Limitations and Directions for Future Research

Several limitations of the present research provide directions for future research. Firstly, the survey population in this study was composed of mHealth app users, and the non-user population was not considered, which may limit the generalizability of the results. This method was appropriate for this research, because some of the constructs explored in this study (e.g., technological characteristics) can only be perceived by users, but the non-user population cannot be ignored. In particular, it is also worth exploring whether non-users’ perceptions of the characteristics of pandemic events affect the adoption of the mHealth app and the subsequent results. Therefore, future research should consider the non-user population to further validate the results of this study. Secondly, when using online surveys, it can be difficult to avoid sampling problems such as self-selection bias, among others. [54,55]. Therefore, we undertook a number of actions to ensure the quality of the survey process and supplemented our findings with qualitative studies. Future research could overcome the sampling bias arising from online surveys through interviews and pre-surveys. Thirdly, mHealth apps can be divided into various types depending on whether their adoption is voluntary (as in the context of this research) or forced (e.g., the Health Code and Travel Code App). While the focus of this paper is on the voluntary adoption setting, future research could examine the effect of forced adoption on users’ app use during public health emergencies.

## Figures and Tables

**Figure 1 ijerph-20-00834-f001:**
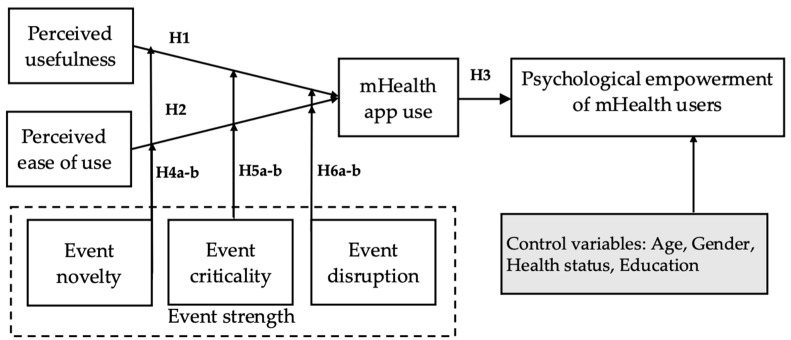
Research Model.

**Figure 2 ijerph-20-00834-f002:**
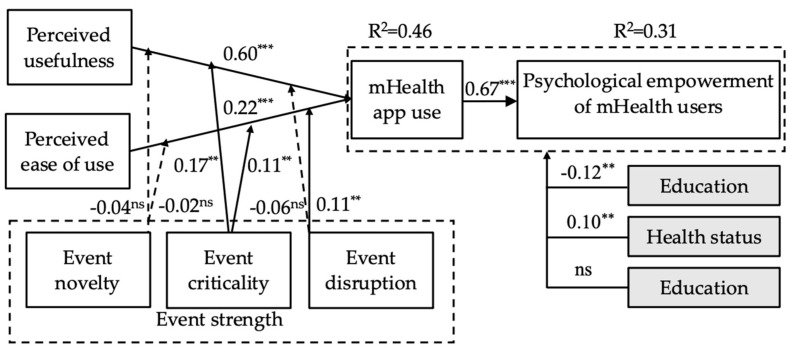
Results of the Research Model Assessment. Note: ** *p* < 0.01, *** *p* < 0.001, ns: not significant.

**Figure 3 ijerph-20-00834-f003:**
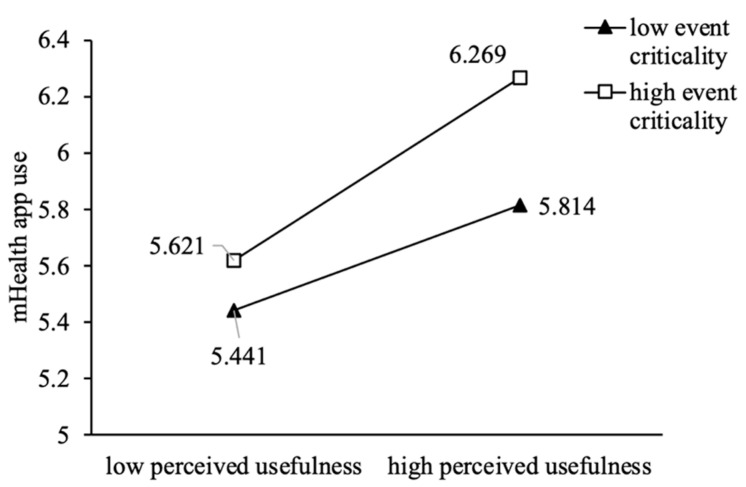
The Moderating Effect of Event Criticality on Perceived Usefulness.

**Figure 4 ijerph-20-00834-f004:**
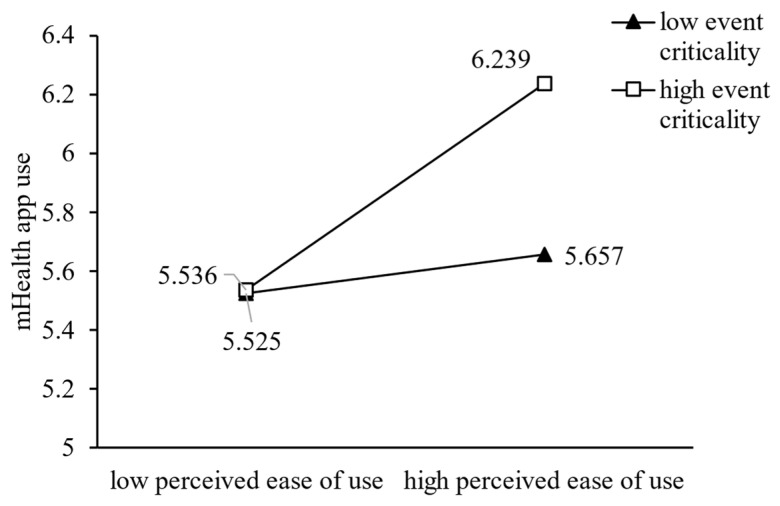
The Moderating Effect of Event Criticality on Perceived Ease of Use.

**Figure 5 ijerph-20-00834-f005:**
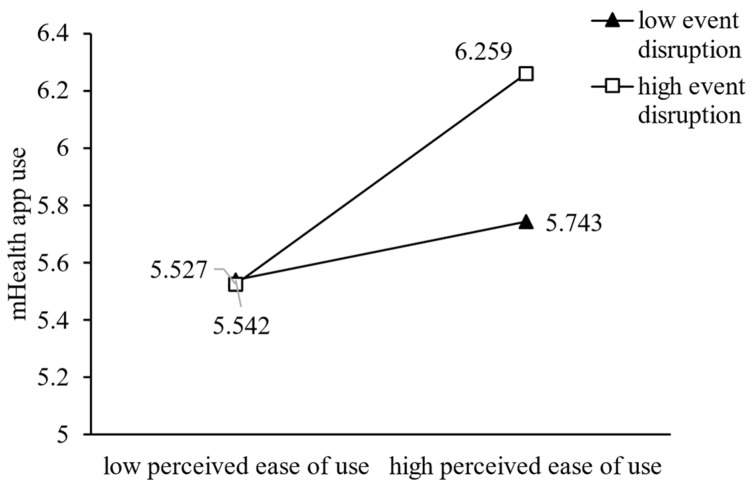
The Moderating Effect of Event Disruption on Perceived Ease of Use.

**Table 1 ijerph-20-00834-t001:** Respondents’ Profiles.

Characteristic	Count	Percentage
Gender		
Male	171	42.5%
Female	231	57.5%
Age		
18–25	156	38.8%
26–35	83	20.6%
36–45	92	22.9%
46–55	49	12.2%
56 or above	22	5.2%
Education		
High school or below	45	11.2%
Junior college	66	16.4%
Undergraduate	165	41.0%
Graduate or above	126	31.4%
Monthly income (CNY)		
Less than 3000	139	34.6%
3001–7000	146	36.3%
7001–10,000	65	16.2%
10001–20,000	41	10.2%
Above 20,000	11	2.7%

**Table 2 ijerph-20-00834-t002:** Reliability and Convergent Validity Test Results.

Variable	Items	Factor Loading	Cronbach’s α	AVE	CR
Perceived Usefulness (PU)	PU1	0.765	0.792	0.563	0.837
	PU2	0.706			
	PU3	0.754			
	PU4	0.773			
Perceived ease of use (PEOU)	PEOU1	0.872	0.886	0.796	0.886
	PEOU2	0.912			
Event disruption (ED)	ED1	0.665	0.787	0.629	0.834
	ED2	0.879			
	ED3	0.820			
Event novelty (EN)	EN1	0.782	0.807	0.586	0.809
	EN2	0.779			
	EN3	0.734			
Event criticality (EC)	EC1	0.710	0.818	0.617	0.828
	EC2	0.850			
mHealth app use (MAU)	MAU1	0.818	0.870	0.627	0.870
	MAU2	0.896			
	MAU3	0.774			
	MAU4	0.767			
Psychological empowerment of mHealth users (PEMU)	PEMU1	0.858	0.916	0.788	0.918
	PEMU2	0.903			
	PEMU3	0.901			

**Table 3 ijerph-20-00834-t003:** Discriminant Validity Test Results.

	Mean	SD	MAU	PEMU	PU	PEOU	EC	ED	EN
MAU	5.810	0.857	0.792						
PEMU	5.682	0.861	0.492	0.888					
PU	5.808	0.801	0.595	0.723	0.750				
PEOU	5.702	0.923	0.438	0.497	0.478	0.892			
EC	5.801	1.030	0.345	0.230	0.396	0.316	0.785		
ED	4.783	1.241	0.303	0.134	0.357	0.094	0.379	0.793	
EN	5.993	0.801	0.276	0.354	0.439	0.451	0.636	0.281	0.766

Note: PU, perceived usefulness; PEOU, perceived ease of use; ED, event disruption; EN, event novelty; EC, event criticality; MAU, mHealth app use; PEMU, psychological empowerment of mHealth users.

**Table 4 ijerph-20-00834-t004:** Results of the Hypothesis Testing.

Hypothesis	Path Description and Direction	Unstandardized Path Coefficient	T Value	Supported?
H1	PU to MAU	β = 0.60 ***	6.78	Yes
H2	PEOU to MAU	β = 0.22 ***	4.07	Yes
H3	MAU to PEMU	β = 0.67 ***	9.85	Yes
H4a	EN×PU to MAU	β = −0.04 ^ns^	−0.54	No
H4b	EN×PEOU to MAU	β = −0.02 ^ns^	−0.35	No
H5a	EC×PU to MAU	β = 0.17 **	2.50	Yes
H5b	EC×PEOU to MAU	β = 0.11 **	2.07	Yes
H6a	ED×PU to MAU	β = −0.06 ^ns^	−1.59	No
H6b	ED×PEOU to MAU	β = 0.11 **	2.50	Yes

Note: (1) PU, perceived usefulness; PEOU, perceived ease of use; ED, event disruption; EN, event novelty; EC, event criticality; MAU, mHealth app use; PEMU, psychological empowerment of mHealth users. (2) ** *p* < 0.01, *** *p* < 0.001, ns: not significant.

**Table 5 ijerph-20-00834-t005:** Hypothesis Testing Results.

Hypothesis	Path Description and Direction	Quantitative Study Results Supported?	Qualitative Study Results Supported?
H1	PU to MAU	Yes	Yes
H2	PEOU to MAU	Yes	Yes
H3	MAU to PEMU	Yes	Yes
H4a	EN*PU to MAU	No	No
H4b	EN*PEOU to MAU	No	No
H5a	EC*PU to MAU	Yes	Yes
H5b	EC*PEOU to MAU	Yes	Yes
H6a	ED*PU to MAU	No	No
H6b	ED*PEOU to MAU	Yes	Yes

Note: PU, perceived usefulness; PEOU, perceived ease of use; ED, event disruption; EN, event novelty; EC, event criticality; MAU, mHealth app use; PEMU, psychological empowerment of mHealth users.

## Data Availability

The data presented in this study are available on request from the corresponding author. The data are not publicly available due to ethical restrictions.

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
