# Peer review of "Understanding the Antecedents and Effects of mHealth App Use in Pandemics: A Sequential Mixed-Method Investigation"

_ijerph, 2023, doi:10.3390/ijerph20010834_

Round 1

Reviewer 1 Report (Previous Reviewer 2)

Dear authors,

thanks for the detailed answer. In my opinion, the differences with the work of Zhou et al. that you indicated in the reply letter to the reviewers should also be indicated, especially for the part the part "in terms of research models, the model of this paper aims to explore the impact of mHealth app characteristics assessment on mHealth app use, contingent on the moderation of pandemic events, and the impact of mHealth app use on psychological empowerment of mHealth users.In comparison, the paper in JASIST models the impact of technology characteristics on psychological empowerment of mHealth users and the impact of pyschological empowerment of mHealth users on subjective well-being, considering the moderation of pandemic events. Indeed, none of the hypotheses in the model of this paper is identical to any hypothesis in the model of the paper in JASIST.", adapting it to the context.

I appreciated the revision of the plagiarized sentences.

Best regards.

Author Response

We appreciate your substantive and constructive comments. Following your suggestion, we have rewritten the introductory section to strengthen the differences from the paper in JASIST. The revisions are as follows (lines 63-65): "Moreover, prior mHealth studies mainly explained the consequences of psychological empowerment of mHealth users[16], lacking in exploring the mechanism of how psychological empowerment of mHealth users are formed ". We hope our revisions and responses are satisfactory to you.

Reviewer 2 Report (Previous Reviewer 1)

I am assuming that the final version will consider all the corrections marked in red.

The paper addresses an interesting topic and it has a consistent scientific structure. Thus, I recommend its publication.

Author Response

Thank you very much for your positive comments.

Reviewer 3 Report (New Reviewer)

The authors present an interesting model for predicting the adoption and outcome of mHealth apps. I have following comments for improving the paper.

INTRODUCTION

- First few sentences (29-39) of the introduction do not fit the rest of the paper. For example, "developing countries" and "limited resources" are never discussed in the remaining paper and their relationship to the model is never discussed. Same thing can be said about "accessibility" and "interaction modality". This part should discuss the broader field of which model is a part of.

- Line 31 - "are vulnerable" is unnecessary.

STUDY 1 - RESULTS

- Authors need to clarify how they controlled for age, gender, health status and education during the data analysis since respondents had different demographic characteristics.

- Figure 3 and Figure 4 - y-axis need to be changed to "event criticality"

STUDY 2 - DATA COLLECTION

- Authors should clarify upfront what type of qualitative data was collected - moreover what kind of articles were identified? academic? blogs? news? 

- It is not clear why the stated platforms were used for searching qualitative data. For example, WeChat is a social media app, how did the authors use this to search for user reviewers and articles.

- Provide a few keywords that were used for searching purposes.

DATA ANALYSIS

difference between first-order and second-order themes should be clarified.

- there are several different types of qualitative data analysis techniques, authors need to clarify which one was used and why it was chosen.

RESULTS

- Authors need to provide their results / findings of qualitative analysis. A table with first-order and second-order theme frequencies should be included. 

It is unclear how the qualitative analysis ended up generating support for H4a-b and H6a.

- There is some confusion about what authors are describing in their results. H4a, b and H6a and b, as described in this paper, do not mention anything about the urgency of medical conditions.

DISCUSSION

- Authors make a lot of claims but do not follow-up on their claims with details and explanations. For example, what do they mean by unilateral in the second implication? Similarly, what mechanisms have been manifested regarding the development and impact of mHealth app use from process perspective. 

- Overall, theoretical implications should be revised to more explicitly clarify the identified mechanisms.

- practical implications do not seem novel. Don't we already know that perceived ease of use and perceived usefulness predict intention to use?

- authors can clarify who might be mHealth operators?

- The research does not provide any insight about improving the designs of mhealth apps.

- Overall,  theoretical and practical implications sections need to be strengthened and the implications should be made more explicit. 

Author Response

Thank you for your favorable reviews and helpful suggestions for improving our manuscript. Please see attached file.

Round 2

Reviewer 3 Report (New Reviewer)

The authors have addressed my comments to some extent. The paper needs to proofreading by a native speaker. 

Author Response

Thank you for your favorable reviews and helpful suggestions for improving our manuscript. Please see attached file.

This manuscript is a resubmission of an earlier submission. The following is a list of the peer review reports and author responses from that submission.

Round 1

Reviewer 1 Report

Since online surveys present some important limitations, I suggest the following readings:

Wright, K. B. (2005). Researching Internet-based populations: Advantages and disadvantages of online survey research, online questionnaire authoring software packages, and web survey services. Journal of computer-mediated communication10(3), JCMC1034.

Andrade, C. (2020). The limitations of online surveys. Indian journal of psychological medicine42(6), 575-576.

These limitations should be recognized in pint 5.3. 

The purpose of the article is well stated, and the research question is clear and appropriate.

The authors included the most relevant literature, and the research model is very clear about how and why the main variables relate to one another.

There is sufficient information about methodological procedures, namely data collection and statistics.

My only concern is related to the sample and the bias introduced by the online survey.

Nonetheless, I recommend the publication of the paper.

Author Response

(The authors gave the same response as above.)

Round 2

Reviewer 2 Report

Dear Authors,

in my previous review I raised the question of the relationship of your manuscript with the article "User empowerment and well-being with mhealth apps during pandemics: A mix-methods investigation in China" recently published in the Journal of the Association for Information Science and Technology (2022) by Zhou; Jin; Hsu and Tang.

I have not received an answer from you in this regard. I note, however, that the reference to the above article has been completely removed from your work. This solution, in my opinion, does not solve the problem raised but, on the contrary, aggravates it. Therefore, I cannot express any opinion on the changes, otherwise not substantial, made to the manuscript. I can only emphasize the seriousness of the question I raised earlier.

Best regards.